# Vitamin C and E Treatment Blunts Sprint Interval Training–Induced Changes in Inflammatory Mediator-, Calcium-, and Mitochondria-Related Signaling in Recreationally Active Elderly Humans

**DOI:** 10.3390/antiox9090879

**Published:** 2020-09-17

**Authors:** Victoria L. Wyckelsma, Tomas Venckunas, Marius Brazaitis, Stefano Gastaldello, Audrius Snieckus, Nerijus Eimantas, Neringa Baranauskiene, Andrejus Subocius, Albertas Skurvydas, Mati Pääsuke, Helena Gapeyeva, Priit Kaasik, Reedik Pääsuke, Jaak Jürimäe, Brigitte A. Graf, Bengt Kayser, Nicolas Place, Daniel C. Andersson, Sigitas Kamandulis, Håkan Westerblad

**Affiliations:** 1Department of Physiology and Pharmacology, Karolinska Institutet, 171 77 Stockholm, Sweden; victoria.wyckelsma@ki.se (V.L.W.); stefano.gastaldello@ki.se (S.G.); daniel.c.andersson@ki.se (D.C.A.); 2Institute of Sport Science and Innovations, Lithuanian Sports University, 44221 Kaunas, Lithuania; tomas.venckunas@lsu.lt (T.V.); marius.brazaitis@lsu.lt (M.B.); audrius.snieckus@lsu.lt (A.S.); nerijus.eimantas@lsu.lt (N.E.); neringa.baranauskiene@lsu.lt (N.B.); andrejus.subocius@lsmuni.lt (A.S.); albertas.skurvydas@lsu.lt (A.S.); sigitas.kamandulis@lsu.lt (S.K.); 3Clinic of Surgery, Republican Hospital of Kaunas, 44249 Kaunas, Lithuania; 4Institute of Sport Sciences and Physiotherapy, University of Tartu, 50090 Tartu, Estonia; mati.paasuke@ut.ee (M.P.); helena.gapeyeva@ut.ee (H.G.); 5Laboratory of Functional Morphology, University of Tartu, 50090 Tartu, Estonia; priit.kaasik@ut.ee; 6Department of Traumatology and Orthopaedics, Tartu University Hospital, 50090 Tartu, Estonia; reedik.paasuke@kliinikum.ee; 7Laboratory of Kinanthropometry; University of Tartu, 50090 Tartu, Estonia; jaak.jurimae@ut.ee; 8Food and Nutrition Department of Health Professions, Faculty of Health, Manchester Metropolitan University, Manchester M1 5GF, UK; b.graf@mmu.ac.uk; 9Institute of Sports Sciences, University of Lausanne, 1015 Lausanne, Switzerland; bengt.kayser@unil.ch (B.K.); nicolas.place@unil.ch (N.P.); 10Cardiology Unit, Heart, Vascular and Neurology Theme, Karolinska University Hospital, 171 77 Stockholm, Sweden

**Keywords:** sprint interval training, high-intensity interval training, aging, endurance exercise, skeletal muscle, antioxidant treatment, reactive oxygen/nitrogen species, calcium, inflammatory mediators

## Abstract

Sprint interval training (SIT) has emerged as a time-efficient training regimen for young individuals. Here, we studied whether SIT is effective also in elderly individuals and whether the training response was affected by treatment with the antioxidants vitamin C and E. Recreationally active elderly (mean age 65) men received either vitamin C (1 g/day) and vitamin E (235 mg/day) or placebo. Training consisted of nine SIT sessions (three sessions/week for three weeks of 4-6 repetitions of 30-s all-out cycling sprints) interposed by 4 min rest. Vastus lateralis muscle biopsies were taken before, 1 h after, and 24 h after the first and last SIT sessions. At the end of the three weeks of training, SIT-induced changes in relative mRNA expression of reactive oxygen/nitrogen species (ROS)- and mitochondria-related proteins, inflammatory mediators, and the sarcoplasmic reticulum Ca^2+^ channel, the ryanodine receptor 1 (RyR1), were blunted in the vitamin treated group. Western blots frequently showed a major (>50%) decrease in the full-length expression of RyR1 24 h after SIT sessions; in the trained state, vitamin treatment seemed to provide protection against this severe RyR1 modification. Power at exhaustion during an incremental cycling test was increased by ~5% at the end of the training period, whereas maximal oxygen uptake remained unchanged; vitamin treatment did not affect these measures. In conclusion, treatment with the antioxidants vitamin C and E blunts SIT-induced cellular signaling in skeletal muscle of elderly individuals, while the present training regimen was too short or too intense for the changes in signaling to be translated into a clear-cut change in physical performance.

## 1. Introduction

Chronic low-grade inflammation is considered a major factor underlying age-related diseases and functional impairments, including declining physical performance [1,2]. This age-related inflammatory state and decline in physical performance is associated with increased circulating levels of both pro- and anti-inflammatory biomarkers, such as tumor necrosis factor α (TNF-α) and interleukins (IL) [3,4]. The aging process and low-grade inflammatory state have been linked to a redox imbalance with excessive production of reactive oxygen/nitrogen species (ROS) relative to the anti-oxidant defense systems [1], although the general concept of oxidative stress as a major cause of detrimental aging processes remains controversial [5].

Inactivity imposes a large and increasing burden on human health, whereas endurance training has numerous health-promoting effects [6]. Intriguingly, the expression of many inflammatory biomarkers associated with the low-grade inflammation in aging also increase in response to an exercise bout [7]. For instance, chronically elevated systemic levels of the cytokine IL-6 are associated with deteriorating muscle function in aging, whereas exercise-induced IL-6 released from skeletal muscle is associated with improved muscle health [8,9]. Along the same lines, the aging-related decline in muscle function has been associated with increased ROS levels; nevertheless, increased ROS production is observed during health promoting endurance exercise [9,10]. In fact, antioxidant treatment has been shown to reduce beneficial effects of endurance training [11,12,13,14], although other studies report no blunting of exercise-induced adaptations by antioxidant treatment [15,16,17].

One frequently used argument against performing regular exercise is that it takes too much time. From this perspective, sprint interval training (SIT), which consists of short bouts of high-intensity exercise (e.g., cycling or running) alternated with recovery periods, has emerged as a popular time-efficient alternative to traditional continuous endurance training at lower intensity [18]. The mechanisms behind muscular adaptations to SIT remain incompletely understood. We recently proposed a central role in the adaptation process of a SIT-induced modification of the sarcoplasmic reticulum (SR) Ca^2+^ release channel, the ryanodine receptor 1 (RyR1) [19]. Leaky RyR1 might lead to a prolonged increase in cytosolic free [Ca^2+^] ([Ca^2+^]_i_) at rest, which could stimulate mitochondrial biogenesis and hence improve fatigue resistance [20,21,22]. We linked SIT-induced RyR1 modification to markedly increased ROS production during energetically demanding high-intensity exercise bouts [19]. Notably, the increase in ROS during SIT is several-fold larger than during fatiguing contractions at lower intensities [19,23,24]. As described above, this massive exercise-induced ROS increase would presumably interact with an already existing increase in ROS-mediated modifications in aged muscle fibers. In general agreement with this reasoning, a recent study on skeletal muscles from aged (22-month-old) mice showed severe RyR1 degradation and this was accompanied by a markedly larger increase in ROS production during repeated tetanic stimulation than in muscles from young (four-month-old) mice [25]. Interestingly, this RyR1 degradation in aged muscles could be prevented by life-long training [25].

In the present study, elderly men performed three weeks of SIT while being treated with either placebo or vitamin C and E, which have been used as an antioxidant treatment in previous training studies [11,12]. We hypothesized that vitamin C and E treatment would blunt SIT-mediated changes in skeletal muscle gene expression of inflammatory mediators as well as of ROS-, Ca^2+^-, and mitochondria-related proteins. In general accordance with our hypotheses, at the end of the three-week training period, we observed blunted SIT-induced changes in mRNA expression in the vitamin C and E–treated group.

## 2. Materials and Methods

### 2.1. Participants and Study Outline

Recreationally active elderly men participated in this double-blind study; none of the participants were engaged in any structured sport training program. Eighteen individuals completed the study and their characteristics are presented in Table 1. Participants were randomly divided into either a vitamin or a placebo group. Vitamins were provided in the form of oral vitamin C (1 g daily) and vitamin E (235 mg daily) [12]. Treatments were initiated seven days before the first SIT session. Vitamin C was taken as 500 mg in the morning and 500 mg in the evening. Each participant took vitamin E either with the morning or the evening meal. Placebo tablets were taken at the same times. On the training days, tablets were taken at least one hour before the training session. The protocol was approved by the regional ethics committee and was in agreement with the Declaration of Helsinki. All participants gave written informed consent before participation.

Training consisted of nine sessions (three sessions/week for three weeks). Each SIT session started with a warm-up consisting of 8 min cycling at a power (W) equal to the individual’s body mass (kg). Then followed 4-6 repetitions of 30-s all-out cycling bouts (Wingate tests) with 4 min of rest between bouts [19]. Sessions 1 and 7–9 were composed of 6 sprints; sessions 2 and 3 were composed of four sprints, and sessions 4–6 were composed of five sprints [26]. A cycle ergometer with continuous power recording was used to quantify the amount of work produced during each SIT session. Muscle function testing using electrical stimulation was performed before and directly (~2 min) after, 1 h after, and 24 h after the first and last SIT sessions. Vastus lateralis muscle biopsies were taken before, 1 h after, and 24 h after the first and last SIT session. Participants were told to maintain their regular diet and no food intake for at least two hours before the first and the last SIT sessions as well as the pre- and post-training testing sessions. Participants were familiarized with the SIT training and experimental procedures for muscle function testing on a separate occasion before the actual testing. Note that due to technical reasons (e.g., limited muscle biopsy material), some analyses could not be performed in all individuals.

Participants visited the laboratory seven days prior to the first SIT session and 48 h after the last SIT session for assessment of maximal power output and maximal oxygen uptake (VO_2max_) using a standard incremental test to exhaustion on a cycle ergometer (Ergometrics 800S, ErgoLine, Medical Measurement Systems, Binz, Germany). Expired gases were measured breath-by-breath with a calibrated mobile testing system (Oxycon Mobile, Jaeger/VIASYS Healthcare, Hoechberg, Germany), and heart rate (HR) was measured with an HR monitor (S625X, Polar Electro, Kempele, Finland). The cycle ergometer test started with 3 min of cycling at 40 W, after which load was increased by 5 W every 10 s. Participants were told to pedal at a cadence of 70 rpm, and the test was continued until they could no longer pedal at this frequency. Maximal values of VO_2_, respiratory exchange ratio (RER), and HR were calculated as the highest average over 20 consecutive seconds.

### 2.2. Muscle Function

Electrically stimulated isometric force of the dominant leg knee extensors (m. quadriceps) was measured using an isokinetic dynamometer (System 3, Biodex Medical Systems, Shirley, MA, USA). The participants were placed in the dynamometer chair with the knee joint adjusted at an angle of 70° (full knee extension = 0°). To minimize the changes in body position, shank, trunk, and shoulders were tightly fastened by belts with Velcro straps. Transcutaneous muscle stimulation was applied via a pair of carbonized rubber electrodes covered on the inner surface with a thin layer of electrode gel (ECG–EEG Gel, Medigel, Modi’in, Israel). One electrode (6 cm × 11 cm) was placed transversely across the width of the proximal segment of the m. quadriceps close to the inguinal ligament, and the other electrode (6 cm × 20 cm) was placed above the distal segment of the m. quadriceps close to the patella. An electrical stimulator (MG 440, Medicor, Budapest, Hungary) was used to deliver 1-ms square-wave supramaximal pulses (voltage 10% higher of that pre-determined to elicit peak torque to a single pulse). Peak torque in response to 1-s trains of pulses given at 20 Hz and 100 Hz delivered at 5-s intervals was measured. Change in the 20 Hz/100 Hz torque ratio was used to evaluate the extent of prolonged low-frequency force depression (PLFFD) [27,28].

### 2.3. Muscle Biopsies

We used previously described and validated procedures for taking needle muscle biopsies [29]. Briefly, after skin sterilization and local anesthesia, a 1–2-mm-long skin cut was made with the tip of a scalpel mid-way over the vastus lateralis muscles of the non-dominant leg using an automatic biopsy device (Bard Biopsy Instrument, Bard Radiology, Covington, GA, USA). A 14-gauge disposable needle was inserted through the cut until the fascia was pierced and the needle was advanced ~2 cm into the muscle, perpendicular to the muscle fibers. Two to three samples (~15 mg each) were collected from one puncture site at each time point, whereas different puncture sites were used for biopsies taken before and 1 h and 24 h after the SIT sessions. A local compression was then applied on the biopsy site for a few minutes to prevent hematoma. Muscle samples were immediately frozen in liquid nitrogen and stored at −80 °C. Due to limited muscle tissue availability, we prioritized measurements of gene expression at 1 h after and protein expression at 24 h after SIT sessions. This prioritization was based on the fact that, after exercise, changes in mRNA levels occur more rapidly than changes in protein content [30], and we have previously observed decreased mRNA levels of exercise-responsive genes 24 h after a SIT session [19].

### 2.4. Protein Analysis

Approximately 15 mg of frozen muscle was weighed and homogenized on ice (1:20 *w*/v) in HEPES lysis buffer (20 mM HEPES, 150 mM NaCl, 5 mM EDTA, 25 mM KF, 5% Glycerol, 1 mM Na_3_VO_4_, 0.5% Triton, pH 7.6) with protease inhibitor (#11836145001, Roche, 1 tablet per 50 mL). Homogenate was diluted to 33 μg wet weight muscle μL^−1^ using 3 × SDS denaturing solution (0.125 M Tris-HCI; 10% glycerol; 4% SDS, 4 M urea; 10% 2-mercaptoethanol; and 0.001% Bromophenol Blue, pH 6.8). Finally, samples were further diluted to 2.5 μg wet weight muscle μL^−1^ with 1 × SDS solution (3 × SDS denaturing solution diluted 2:1 with 1 × Tris.Cl (pH 6.8)). A small amount of homogenate was taken from each participant pre-training to make a calibration curve to be run on every gel for Western blotting. Homogenates were stored at −80 °C until analysis.

Protein was separated on 26-well 4–15% TGX stain-free gels or 26-well 4–12% Bis-Tris Gels (Bio-Rad). TGX stain-free gels had total protein visualized prior to transfer and analysis on Image Lab software (Image Lab 6.0, Bio-Rad). Protein was wet transferred to PVDF membrane at 100 V for 1 h (TGX) or 90 min (Bis-Tris). Following transfer, membranes were blocked at room temperature for 2 h using LI-COR blocking buffer with PBS (LI-COR Biosciences, Lincoln, NE, USA). Additionally, following transfer, Bis-Tris gels were Coomassie stained (Coomassie Brilliant Blue R-250, Bio-Rad) for 2 h at room temperature, then de-stained (40% Methanol, 10% Acetic Acid) for 2 × 1 h washes at room temperature, and then overnight in MilliQ H_2_O before being visualized for myosin bands on a Chemi Doc MP (Biorad). After blocking, membranes were incubated in primary antibody overnight at 4 °C and for 2 h at room temperature. Primary antibody details are as follows: RyR1 (1:100, mouse, 34C, Developmental Studies Hybridoma Bank (DHSB), University of Iowa, IA, USA), SR Ca^2+^-ATPase 1 (SERCA1; 1:1000, mouse, CaF2-5D2, DHSB), SERCA2a (1:5000, rabbit, A010-20, Badrilla), calsequestrin 1 (CSQ1; 1:1000, mouse, MA3-913, Thermofisher), CSQ2 (1:1000, rabbit, ab3516, Abcam), and total OXPHOS antibody cocktail (1:1000, mouse, ab110413, Abcam). All antibodies were diluted in LI-COR Blocking Buffer in PBS (LI-COR Biosciences) 1:1 *v*/*v* with 1× TBST. After incubation in primary antibody, membranes were washed in 1x TBST, incubated in secondary antibody (1:10,000, IRDye 680-conjugated donkey anti-mouse IgG and IRDye 800-conjugated donkey anti-rabbit IgG (926–68,072, 926–32,213, LI-COR Biosciences), and immunoreactive bands were visualized using infrared fluorescence on an IR-Odyssey scanner. Band densities were analyzed using Image Studio Lite v 5.2 (LI-COR Biosciences). During data analysis, the density of each given protein was measured relative to the calibration curve and then normalized to the total protein as measured for each lane in stain-free gels or Coomassie-stained gels for RyR1. The same calibration curve was used across all gels and data are expressed relative to the average of the pre training biopsies on each gel.

### 2.5. Gene Expression Analysis

Total RNA was isolated from muscle biopsies using TRIzol Reagent solution (15596026, Thermo Fisher Scientific, Vilnius, Lithuania) following the manufacture instructions. Extracted RNAs were purified from DNA contamination with DNase I treatment (EN0521, Thermo Fisher Scientific, Lithuania) for 1 h at 37 °C. The correspondent cDNAs were produced using both oligo (dT)18 and random primers by following the instruction of RevertAID H Minus First strand cDNA synthesis Kit (K1632, Thermo Fisher Scientific, Lithuania). Quantitative real-time PCR were performed with 100 ng of cDNA template using SYBR Green Master Mix (A25741, Life Technologies, Carlsbad, CA, USA) in a final volume of 20 μL. The essay was performed with QuantStudio 3 Real-Time PCR Systems (Thermo Fisher Scientific, Waltham, MA, USA) using the following cycling program: initial 50 °C 2 min, denaturation 95 °C 10 min, followed by 40 cycles of 95 °C for 15 s and 60 °C for 1 min. A final step of melting curve between 65 °C and 90 °C, 1 °C/sec temperature speed was incorporated. Table 2 shows the primers used with a stock concentration of 10 µM. Hypoxanthine-guanine phosphoribosyltransferase (HPRT) was used as the housekeeping control gene, which did not change with any intervention. Fold change relative to the gene expression was calculated as 2^−ΔΔCt^, where ΔΔC_t_ = ΔC_t_ (target gene 1 h after SIT session)-ΔC_t_ (target gene before SIT session). ∆C_t_ = C_t_[Target]-C_t_[Housekeeping], according to the Minimum Information for Publication of Quantitative Real-Time PCR Experiments (MIQE) guideline. All single samples were analyzed in triplicate, and the mean value was used in subsequent analyses. Gene expression data obtained before the first and the last SIT sessions (i.e., in the untrained and trained state, respectively) in individual participants are presented in Appendix A (Appendix A).

### 2.6. Blood Lactate Measurement

Blood lactate concentration was measured in fingertip blood samples taken before and 5 min and 60 min after the first and last SIT sessions using a portable lactate analyzing unit (ProTM LT-1730, Arkray Inc., Kyoto, Japan).

### 2.7. Statistical Analysis

Statistical analyses were performed with SigmaPlot software (v11; Systat, Chicago, IL, USA). Two-way repeated measures ANOVA was used to identify general group and treatment differences between vitamin- and placebo-treated participants. When the ANOVA showed a significant effect, the Holm–Sidak post hoc test was used to identify differences in the expression of individual genes or proteins between the vitamin- and placebo-treated participants. The α level for statistical significance was set to *p* < 0.05. Data are reported as mean ± SD and as individual values.

## 3. Results

### 3.1. Performance during SIT Sessions

The mean power output during the individual Wingate cycling bouts was measured during the first and last SIT sessions in some individuals in each group. The results show only a minor decline in mean power over the six cycling bouts in the untrained (first SIT session) and trained (last SIT session) states as well as in the vitamin and the placebo groups (Appendix A, Appendix A). The mean power over the six Wingate cycling bouts was slightly (~7%) larger in the trained than in the untrained state, but the difference did not reach statistical significance with the limited number of individuals tested (*n* = 4 in each group) (Appendix A). Possible training effects on maximal cycling performance were assessed in the same subgroup of individuals by comparing the peak and mean power during the first Wingate cycling bout in the untrained and trained state [31]. Our results show no consistent training effect on these measures of power output (Appendix A).

Blood lactate was measured in some individuals and showed an expected increased to ~15 mM at 5 min after both the first and last SIT sessions in both the vitamin and the placebo groups (Appendix A). Thus, from this perspective, the energy metabolic stress during the SIT sessions was similar in all these four instances.

### 3.2. Gene Expression of Inflammatory Mediators

We measured the mRNA expressed by a battery of genes encoding for inflammatory mediators, which have been shown to increase in muscle in response to exercise [7], plus the nonhistone nuclear protein alarmin HMGB1, which is expressed in the myoplasm of adult muscle fibers under inflammatory conditions [32] (Figure 1). The seven days of vitamin treatment before the first SIT session resulted in a lower mRNA expression (relative to the house-keeping gene) of IL-1β than in the placebo group, whereas the expression of the other genes tested did not differ between the two groups (Appendix A, Appendix A). The first SIT session (i.e., untrained state) did not result in any statistically significant difference in the relative mRNA expression of inflammatory mediators at the group level (i.e., the vitamin- vs. the placebo-treated group), although the relative expression of IL-1β and IL-10 were larger in the vitamin group and IL-6 in the placebo group. In trained state, on the other hand, there were highly significant general differences between the vitamin and placebo group (*p* < 0.001) both for the genes showing the largest SIT-induced increases (IL-1β and IL-6, Figure 1a) and those showing more moderate effects (Figure 1b). Note that in the trained state, the relative expression was larger for all genes tested in the placebo group than in the vitamin group, which indicates blunting of the exercise-induced inflammatory response in individuals treated with the antioxidants vitamin C and E.

### 3.3. Gene Expression of ROS-Related Proteins

The production of ROS has been shown to be markedly increased during high-intensity exercise [19], and we hypothesized that SIT-induced changes in the transcription of genes encoding for ROS-related proteins would be blunted by the vitamin treatment. In general agreement with the inflammatory mediators´ results, we observed a difference in exercise-induced changes in the relative mRNA expression of ROS-related proteins at the group level in the trained state (*p* < 0.05) but not in the untrained state. Intriguingly, at the single gene level the relative NOX4 mRNA expression was larger in the vitamin group in the untrained state, whereas it was larger in the placebo group in the trained state (Figure 2).

Assessment of mRNA expression (relative to the house-keeping gene) before SIT sessions showed a higher expression of SOD1 in the placebo than in the vitamin group in both the untrained and trained states (Appendix A, Appendix A). Moreover, we observed a training-induced increase in the mRNA expression of NOX2 before SIT sessions in both the vitamin and the placebo groups.

### 3.4. Protein and Gene Expression of SR Ca^2+^-Handling Proteins

#### 3.4.1. Full-Length RyR1 Protein Expression

We have previously shown that SIT sessions can induce modifications in the RyR1 resulting in a reduction in full-length protein expression in Western blots [19,26]. Therefore, we assessed the full-length RyR1 expression in muscle biopsies taken before and 24 h after the first (untrained state) and last (trained state) SIT sessions (Figure 3. The full-length RyR1 expression was significantly (*p* < 0.05) decreased 24 h after the first SIT session in both groups (Figure 3b). Conversely, full-length RyR1 expression was not significantly decreased 24 h after the last SIT session, and the results showed a marked variability at this time point. At the individual participant level in the trained state, we observed a major SIT-induced reduction in full length RyR1 expression to <50% of the pre-exercise value in three out of four placebo -treated participants compared to one out of five participants in the vitamin group (see Figure 3b). Thus, this assessment at the individual level might be taken in support for some vitamin-mediated protection against RyR1 modification.

#### 3.4.2. Gene Expression of RyR1

We also assessed the relative mRNA expression one hour after SIT sessions in the untrained and trained state using primers directed either to the C- or N-terminal of RyR1 (Figure 3c). In agreement with measurements of mRNAs for inflammatory mediators and for ROS-related proteins, we observed a difference in SIT-induced changes in RyR1 mRNA expression at the group level in the trained state (*p* < 0.01) but not in the untrained state. After the last SIT session, the relative RyR1 mRNA expression was significantly larger in the placebo group than in the vitamin group. Moreover, in the vitamin group we observed a lower mRNA expression (relative to the house-keeping gene) of RyR1 before SIT sessions in the trained than in the untrained state (Appendix A, Appendix A).

#### 3.4.3. Gene and Protein Expression of SR Ca^2+^-Handling Proteins Other Than RyR1

We also assessed whether the three weeks of SIT affected the mRNA and protein expression of other SR Ca^2+^-handling proteins, i.e., the Ca^2+^-pumps SERCA1 and 2a, the Ca^2+^-buffers calsequestrin (CSQ) 1 and 2, and, at the mRNA level, the t-tubular voltage sensor DHPR (Figure 4). At the mRNA level, we observed a significant general group difference in the untrained state (*p* < 0.01), which was accompanied by a markedly larger relative mRNA expression of CSQ2 in the placebo than in the vitamin group. In the trained state, we detected no general group difference, although the relative mRNA expression of SERCA1 and CSQ1 were larger in the placebo than in the vitamin group. At the protein level, we observed no significant training-induced changes in the expression of these proteins or differences between vitamin- and placebo-treated participants.

Assessment of mRNA expression (relative to the house-keeping gene) before SIT sessions showed a higher expression of CSQ1 in the vitamin than in the placebo group in both the untrained and trained states (Appendix A, Appendix A). Furthermore, we observed a training-induced decrease in the mRNA expression of DHPR before SIT sessions in the vitamin group.

### 3.5. Aerobic Capacity

#### 3.5.1. Gene and Protein Expression of Mitochondria-Related Proteins

The acute SIT effects on mitochondria-related gene expression were assessed by measuring mRNA levels before and one hour after a SIT session performed in the untrained and trained states. In the untrained state, the results showed no significant difference between the two groups in SIT-induced changes in mRNA expression of mitochondria-related proteins. In the trained state, on the other hand, there was clear overall group difference (*p* < 0.001) with significantly larger relative mRNA expression for five out of seven measured genes in the placebo than in the vitamin group (Figure 5a). Measurements of training-induced changes in protein expression in the five mitochondrial electron transport chain complexes did not reveal any significant training effects or differences between vitamin and placebo-treated individuals, although there was a trend for generally lower relative protein expressions in the trained state in the vitamin than in the placebo-treated participants (*p* = 0.15) (Figure 5b,c).

Measurements of mRNA expression (relative to the house-keeping gene) before SIT sessions showed a higher expression of MUL1 in the placebo than in the vitamin group in both the untrained and trained states (Appendix A, Appendix A). Moreover, we observed a training-induced decrease in the mRNA expression of MFN2 before SIT sessions in the vitamin group.

#### 3.5.2. Incremental Cycling Test to Assess VO_2max_

Incremental cycling tests performed before the training period revealed no notable difference between individuals in the vitamin- and placebo-treated groups regarding VO_2max_, maximal cycling power, maximal heart rate, or maximal respiratory exchange ratio (RER_max_) (Table 1). After the three weeks of SIT, there was a minor, but significant, improvement in power output at exhaustion, and the magnitude of this improvement was similar in the vitamin (4.3 ± 4.0%) and the placebo (6.8 ± 7.6%) groups (Figure 6a). Still, we observed no significant increase in VO_2max_ after the three weeks of SIT in individuals treated either with vitamins (relative increase 2.5 ± 7.7%) or placebo (relative increase 1.2 ± 7.7%) (Figure 6b). Likewise, there was no significant difference between the untrained and trained state in either the vitamin or the placebo group regarding the maximal heart rate (relative increase 6.0 ± 12.3% and 0.4 ± 3.1%) or RER_max_ (relative increase 5.4 ± 5.7% and 1.1 ± 10.7%).

### 3.6. Recovery of Isometric Force after SIT Sessions

In young individuals, we previously observed a prolonged depression of electrically evoked isometric knee extension force after six repeated 30 s Wingate cycling bouts, which was more marked at low than at high stimulation frequencies, i.e., muscles entered a state of prolonged low-frequency force depression (PLFFD) [19]. We here observed decreased electrically evoked torques in quadriceps muscles one hour after SIT sessions, with no significant differences between vitamin- and placebo-treated participants (Figure 7). The torque evoked at 20 Hz was more depressed than at 100 Hz in the untrained state, hence PLFFD was present. There was clear tendency for torque recovery to be improved in the trained state in both groups; for instance, 20 Hz torque was decreased by ~50% in the untrained state and by ~30% in the trained state at one hour of recovery.

## 4. Discussion

We here tested the hypothesis that treatment with the antioxidants vitamin C and E blunts the response to three weeks of SIT in elderly recreationally active individuals. Accordingly, we observed a general blunting of changes in mRNA expression of ROS-related, inflammatory, and mitochondria proteins induced during the last SIT session in vitamin-treated as compared to placebo-treated individuals. On the other hand, the effect of three weeks of SIT on exercise performance measures was similarly small, or absent, in vitamin- and placebo-treated individuals.

Previous studies have shown attenuated beneficial muscle adaptations to endurance training in humans exposed to the same vitamin C and E treatment as in the present study [11,12]. Still, in the study of Paulsen et al. [12], the blunted increase in markers of mitochondrial biogenesis in trained muscle of vitamin-treated individuals was not accompanied by diminished improvements of VO_2 max_ and running performance. This lack of vitamin effects on aerobic performance is in accordance with the present results, where vitamin treatment blunted training-induced changes in mRNA expressions, while VO_2max_ and cycling performance were not affected.

Endurance exercise triggers transient and repeated increases in mRNA expression that drive changes in muscle protein content and ultimately result in training-induced adaptations. Thus, after an exercise session, changes in mRNA expressions occur on a minutes-hours timescale, whereas changes in protein content normally develops over days to weeks [33]. We observed little difference in the measured mRNA expressions between rested muscle (i.e., before SIT sessions) in the untrained and the trained state (see Appendix A), which would imply that the changes in mRNA expression observed after SIT sessions typically returned to baseline before the next SIT session. In order to limit the number and size of muscle biopsies from each participant, measurements of mRNA after exercise were only performed on biopsies taken one hour after the end of SIT sessions. The one-hour post-exercise time point was chosen so as not to miss short-lasting peaks in mRNA expression, but it was likely too early for peak expression of most genes [30,34]. Still, a consistent pattern of mRNA expression changes was detected one hour after the last SIT session, even though the measured changes may not represent the peak amplitudes.

We observed limited and inconsistent differences between the vitamin and placebo groups in mRNA expression before SIT sessions, which indicates that vitamin treatment had little effect on baseline gene expression. Intriguingly, differences between the two groups in the response to the first SIT session did not show any clear pattern, whereas there was distinct pattern of blunted mRNA expressions in the vitamin group after the last SIT session. Further experiments are required to establish the mechanisms underlying this delayed effect of vitamin treatment on the response to exercise.

### 4.1. Effectiveness of SIT in Young vs. Elderly Individuals

Various types of high-intensity interval training have emerged as effective and time-efficient endurance training regimes for healthy individuals and individuals suffering from diseases with severe negative impact on endurance exercise capacity, such as heart failure, metabolic syndrome, and cancer [18,35,36,37,38]. Studies on young (~20–30 years old) sedentary and recreationally active individuals exposed to a period of SIT similar to that used in the present study report increases in peak and mean power during all-out Wingate cycling as well as improved performance during more long-lasting tests depending on aerobic capacity [39,40,41]. In our elderly individuals, the effects of SIT were less pronounced. Even though maximal power output at exhaustion in the incremental cycling test was increased by ~5%, this was not accompanied by any increase in VO_2max_. Moreover, the three weeks of SIT did not result in any significant improvement in cycling performance during Wingate tests.

A high VO_2max_ implies good aerobic fitness and is associated with good health, whereas low VO_2max_ is associated with several common and life-threatening diseases [42]. Thus, physical activities that increase VO_2max_ have clear health-promoting effects. Recent meta-analyses show increased VO_2max_ in young sedentary or recreationally active individuals after a period of high-intensity interval training similar to that used in the present study [31,43,44]. An increased VO_2max_ was also observed in elderly (mean age: 63 years) individuals after nine sessions of high-intensity cycling bouts comprising 6 × 30 s sprints interspersed by 3 min of rest [45]. Intriguingly, our results reveal no SIT-induced increase in VO_2max_, even though our participants also performed nine sessions of virtually the same SIT sessions as in Knowles et al. [45]. The finding that our training regimen was ineffective in improving muscle aerobic capacity is supported by the fact that we did not detect any significant training-induced increase in mitochondrial protein expression. At the onset of the training period, the sedentary individuals in the study of Knowles et al. [45] had somewhat lower VO_2max_ than our recreationally active individuals (average 27 vs. 33 mL kg^−1^ min^−1^). However, this difference in initial aerobic fitness is unlikely to explain the lack of training response in our study, since Knowles et al. also observed a training-induced increase in VO_2max_ in lifelong exercisers having a mean VO_2max_ of 39 mL kg^−1^ min^−1^ at the start of the training period [45]. On the other hand, an important difference was that participants performed the nine SIT sessions over three weeks in our present study and over six weeks in Knowles et al. [45]. The shorter recovery period in our study might limit the training response, and a previous study has shown that, in young individuals, three days are enough for full recovery peak cycling power following a single SIT session, whereas five days are required for full recovery in elderly individuals [46]. Thus, the recovery time between SIT sessions (2–3 days) in our study might have been too short to allow for an increase in aerobic capacity. Alternatively, three weeks of training might be too short for a detectable increase in VO_2max_ to be established.

### 4.2. Expression of Genes Encoding for Inflammatory Mediators

Our muscle biopsy analyses revealed markedly larger relative mRNA expression of inflammatory mediators after the last SIT session in the placebo than in the vitamin group. This indicates that muscles in the placebo-treated individuals were still in an adaptive phase where SIT caused a major cellular stress at the end of the three weeks of training. Two opposite outcomes of this difference between the two groups can be envisaged: (i) positive effects on muscle exercise performance would be observed in the placebo group after a longer period of the present intense SIT regimen and these improvements would be blunted in the vitamin group or (ii) the present training program was too intensive for the elderly participants and continued SIT-induced increases in mRNA of inflammatory mediators in the placebo group might lead to muscular overtraining and decreased physical performance, which then would be prevented by the vitamin treatment [7,8,47,48].

### 4.3. SR Ca^2+^-Handling

Based on results from mouse muscle experiments, RyR1 modifications potentially resulting in increased SR Ca^2+^ leak and increased [Ca^2+^]_i_ at rest constitute a potent trigger for mitochondrial biogenesis and adaptations towards improved muscular endurance [20,49]. Exercise-induced RyR1 modifications have been associated with increased ROS, and high-intensity exercise represents a strong trigger of ROS production [19,23,24]. On the assumption that the effects of treatment with vitamins C and E can be ascribed to their role as antioxidants, the present results provide several pieces of evidence in support of SIT-induced ROS-dependent effects on RyR1 and SR Ca^2+^ handling in the trained state:(i)Vitamin treatment gave some protection against the exercise-induced decrease in full-length RyR1 expression, as judged from a reduction to <50% of the pre-exercise value after the last SIT session in only one of five vitamin-treated individual as compared to three of four placebo-treated individuals.(ii)The relative RyR1 mRNA expression after the last SIT session was markedly smaller in the vitamin than in the placebo group, which indicates a reduced demand for synthesis of new RyR1 protein with vitamin treatment.(iii)The RyR1 mRNA expression before SIT sessions was lower in the trained than in the untrained state in the vitamin group. This, combined with an almost unaffected full-size RyR1 protein expression before the last SIT session, again indicates a reduced RyR1 protein turnover in trained vitamin-treated participants.(iv)After the last SIT session, the relative mRNA expression was higher for NOX2 and smaller for NOX4 in the vitamin than in the placebo group. A recent study on genetically and pharmacologically manipulated adult mouse skeletal muscle revealed a reciprocal interaction between NOX2 and NOX4 expression, where a decrease in NOX2 was accompanied by increases in NOX4 expression and RyR1-mediated SR Ca^2+^ leak [50]. Thus, it can be speculated that the vitamin treatment counteracted this tentative SIT-induced SR Ca^2+^ leak-promoting shift in the relation between NOX2 and NOX4 protein concentration.

### 4.4. PLFFD

In the pre-training state, the recovery of isometric torque evoked by direct electrical stimulation of the quadriceps muscle after a SIT session was slow and a marked PLFFD (i.e., decreased 20/100 Hz force ratio) was present one hour after exercise in both groups. PLFFD can, in principle, be due to decreased SR Ca^2+^ release and/or reduced myofibrillar Ca^2+^ sensitivity, and increased ROS production has been associated to the development of PLFFD [23,51]. Previous studies from our group have shown that there is no direct coupling between the decrease in full-length RyR1 expression in Western blots and PLFFD after an SIT session: (i) in recreationally active individuals, the RyR1 degradation was maximal at 24 h after exercise, while force had recovered at that time [19]; (ii) PLFFD was present in elite athletes despite no signs of RyR1 fragmentation [19]; (iii) in young recreationally active individuals, three weeks of SIT (same training protocol as in the present study) resulted in some protection against the SIT-induced RyR1 modification, whereas force recovery was not improved [26]. Accordingly, in elderly recreationally active individuals, we here show slightly improved recovery of electrically evoked torques in the trained as compared to the untrained state, whereas this was accompanied by some RyR1 protection only in the vitamin group. Thus, further experiments are required to reveal mechanisms underlying the development of PLFFD after a SIT session and their ROS and Ca^2+^ dependency.

## 5. Conclusions

Our results fit with a model where increased ROS have a key signaling role in the skeletal muscle response to a period of SIT performed by elderly humans. In this model, ROS acted on inflammatory Ca^2+^- and mitochondria-related signaling, which was blunted by treatment with the antioxidant’s vitamin C and E. In contrast to previous studies on younger individuals, the present training protocol (three SIT sessions/week for three weeks) did not result in any clear-cut functional improvements in our elderly participants. Thus, the training period and/or the recovery time between SIT sessions were too short for detectable improvements in aerobic capacity and cycling performance to develop.

## Figures and Tables

**Figure 1 antioxidants-09-00879-f001:**
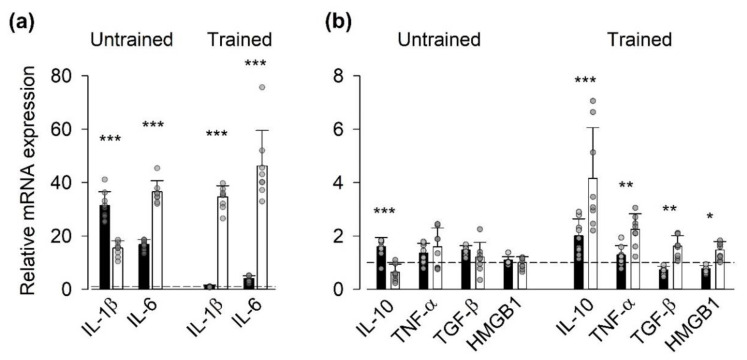
Vitamin treatment blunted the exercise-induced increase in mRNA expression of genes encoding for inflammatory mediators in the trained state. Data (mean ± SD) are expressed as the ratio of mRNA expression 1 h after to that before the first (Untrained) and last (Trained) SIT sessions. Genes showing large exercise-induced effects (**a**) and more moderate effects (**b**). Vitamin group (black bars, *n* = 9); placebo group (white bars, *n* = 7); dashed lines indicate no difference between before and after exercise (i.e., relative mRNA expression = 1). Data from each participant are shown as grey circles. Two-way repeated measures ANOVA showed a general group difference (*p* < 0.001) in the trained state both in (**a**) and (**b**). Difference between vitamin and placebo groups in relative mRNA expression of individual genes: * *p* < 0.05, ** *p* < 0.01, *** *p* < 0.001.

**Figure 2 antioxidants-09-00879-f002:**
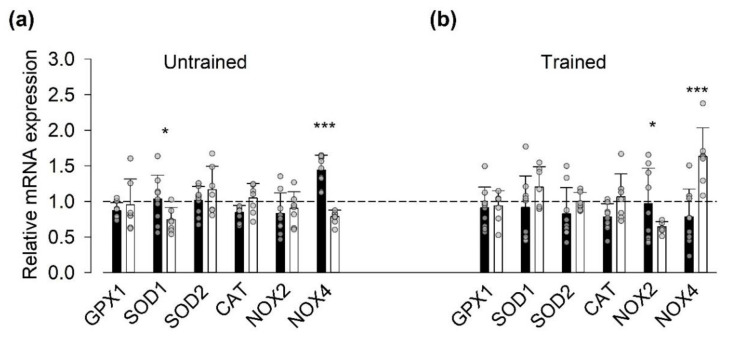
Vitamin-treatment-blunted sprint interval training (SIT)-induced changes in mRNA expression of genes encoding for reactive oxygen/nitrogen species (ROS)-related proteins in the trained state. Data (mean ± SD) are expressed as the ratio of mRNA expression 1 h after to that before the first ((**a**), Untrained) and last ((**b**), Trained) SIT sessions. Vitamin group (black bars, *n* = 9); placebo group (white bars, *n* = 7); dashed line indicates no difference between before and after exercise. Data from each participant are shown as grey circles. Two-way repeated measures ANOVA showed a general group difference (*p* < 0.05) in the trained state. Difference between vitamin and placebo groups in relative mRNA expression of individual genes: * *p* > 0.05, *** *p* < 0.001.

**Figure 3 antioxidants-09-00879-f003:**
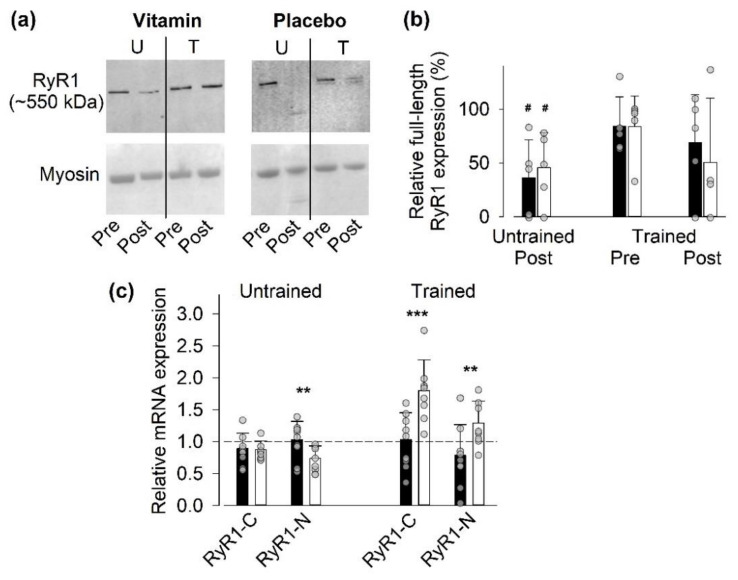
Vitamin-treatment-blunted SIT-induced changes in RyR1 mRNA expression in the trained state. (**a**) Representative Western blots obtained in one vitamin- and one placebo-treated individual before (Pre) and 24 h after (Post) the first (untrained; U) and last (trained; T) SIT sessions. Note the decrease in full-length RyR1 expression in the untrained state in the vitamin-treated participant and a decrease in both the untrained and trained states in the placebo-treated participant. Lower panels show the myosin band in Coomassie-stained gels, which were used as loading controls. (**b**) The relative full-length RyR1 expression in individuals treated with vitamins (*n* = 5) or placebo (*n* = 4). Data are expressed relative to the expression before the first SIT session, which was set to 100% in each subject. No significant difference between the vitamin and placebo groups at any time with two-way repeated measures ANOVA. # denotes significant difference (*p* < 0.05) from the untrained pre-exercise value. (**c**) The ratio of mRNA expression 1 h after to that before the first and last SIT sessions in the vitamin group (*n* = 9) and placebo group (*n* = 7); dashed line indicates no difference between before and after exercise. Two-way repeated measures ANOVA showed a general group difference (*p* < 0.01) in the trained state. Difference between vitamin and placebo groups in relative mRNA expression of individual genes: ** *p* < 0.01, *** *p* < 0.001. Data are shown as mean ± SD; vitamin group, black bars; placebo group, white bars. Data from individual participants are shown as grey circles.

**Figure 4 antioxidants-09-00879-f004:**
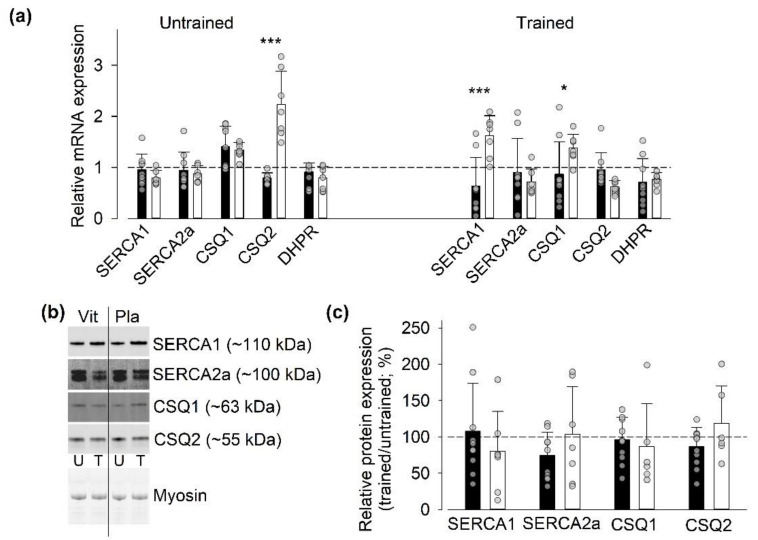
Limited SIT-induced changes in mRNA and protein expression of other SR Ca^2+^-handling proteins than RyR1. (**a**) Mean and individual data of the ratio of mRNA expression 1 h after to that before the first (Untrained) and last (Trained) SIT sessions in individuals treated with vitamins (*n* = 9) or with placebo (*n* = 7); dashed line indicates no difference between before and after exercise. Two-way repeated measures ANOVA showed a general group difference (*p* < 0.01) in the untrained state. Difference between vitamins and placebo groups in relative mRNA expression of individual genes: * *p* < 0.05, *** *p* < 0.001. (**b**) Representative original Western blots from one individual treated with vitamins (Vit) and one treated with placebo (Pla). Biopsies taken before the first (untrained, U) and the last (trained, T) SIT session. Lower panels show stain-free loading controls. (**c**) Mean and individual data of the relative protein expression in biopsies obtained in the untrained and the trained state in individuals treated with vitamins (n = 10) or with placebo (*n* = 6-7). Data expressed as the ratio of the expression in the trained to that in the untrained state; dashed line indicate no difference between the two states. No significant differences between the untrained and the trained state or between vitamin and placebo-treated individuals were observed with two-way repeated measures ANOVA. Data are shown as mean ± SD; vitamin group, black bars; placebo group, white bars.

**Figure 5 antioxidants-09-00879-f005:**
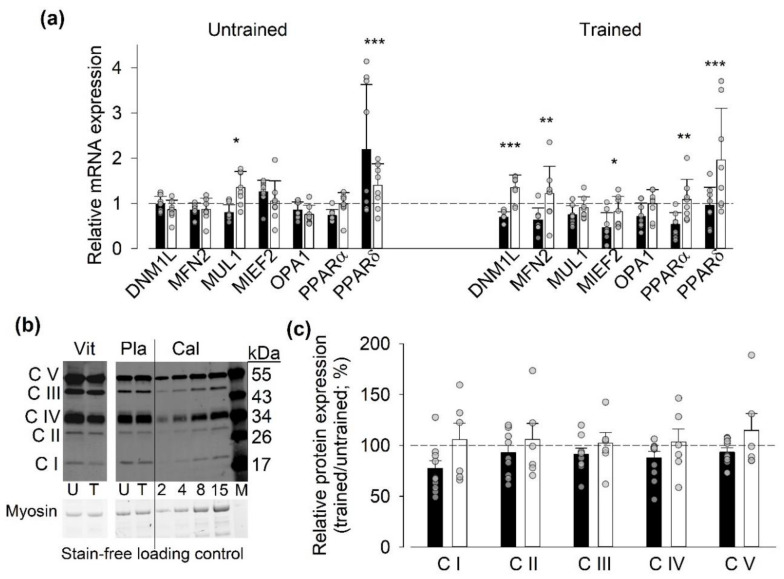
Vitamin-treatment-blunted SIT-induced changes in mRNA expression of genes encoding for mitochondria-related proteins in the trained state. (**a**) Mean and individual data of the ratio of mRNA expression 1 h after to that before the first (Untrained) and last (Trained) SIT sessions in individuals treated with vitamins (*n* = 9) or with placebo (*n* = 7); dashed line indicates no difference between before and after exercise. Two-way repeated measures ANOVA showed a general group difference (*p* < 0.01) in the trained state. Difference between vitamin and placebo groups in relative mRNA expression of individual genes: * *p* < 0.05, ** *p* < 0.01, *** *p* < 0.001. (**b**) Representative original Western blots of mitochondrial complex I–V components (CI-CV) in muscle biopsies from one vitamin (Vit) and one placebo-treated (Pla) individual. Biopsies taken before the first (untrained, U) and the last (trained, T) SIT sessions. Calibration (Cal) shows loading of 2, 4, 8, and 15 µl, which represent 5, 10, 20, and 37.5 µg muscle wet weight. Lane labeled M shows protein size ladder. (**c**) Mean and individual data of the relative protein expression in biopsies obtained in the untrained and trained state in individuals treated with vitamins (*n* = 9-10) or with placebo (*n* = 6). Data expressed as the ratio of the expression in the trained to that in the untrained state. No significant differences between the untrained and trained state or between vitamin- and placebo-treated participants were observed with two-way repeated measures ANOVA. Data are shown as mean ± SD; vitamin group, black bars; placebo group, white bars.

**Figure 6 antioxidants-09-00879-f006:**
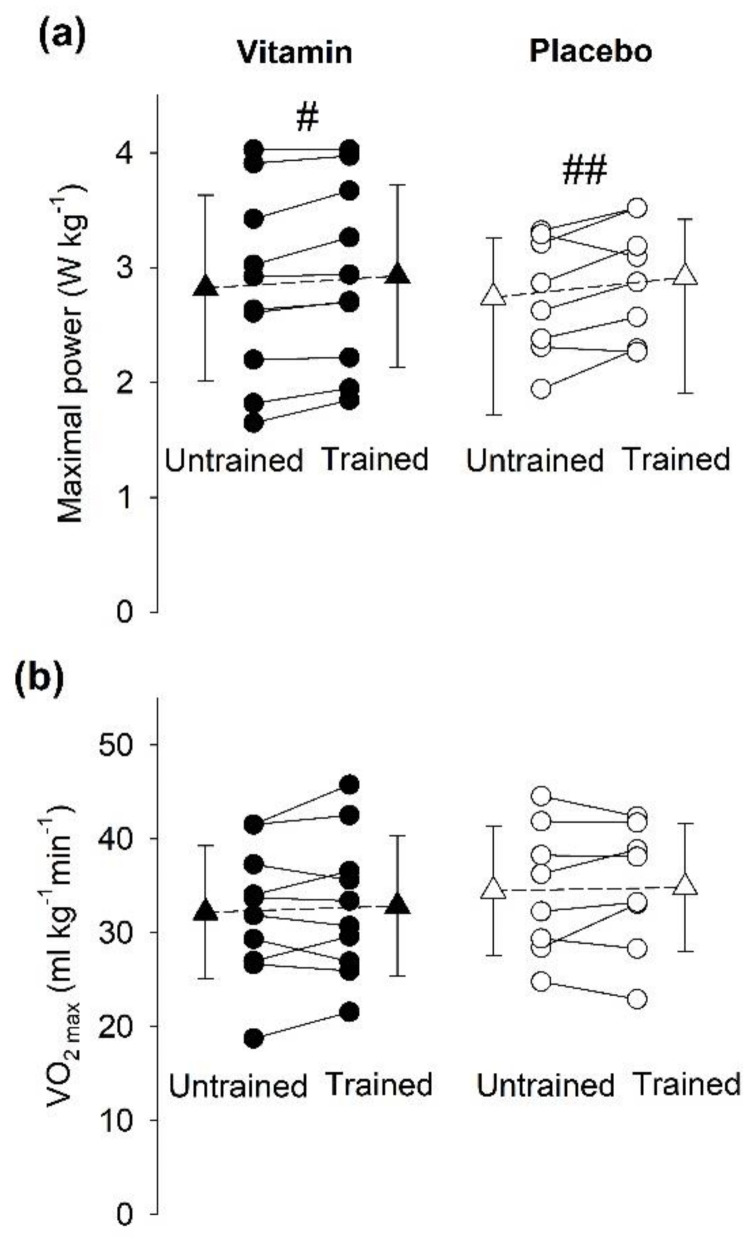
Three weeks of SIT increased power at exhaustion but not VO_2max_. (**a**) Maximal power reached at task failure and (**b**) VO_2max_ during an incremental cycling exercise test prior to the first SIT session (Untrained) and two days after the last SIT session (Trained). Individuals were either treated with vitamins (black symbols; *n* = 10) or placebo (white symbols; *n* = 8). Data from each individual (●,○) and mean data (±SD; ▲,Δ). Two-way repeated measures ANOVA showed a significant training effect for maximal power in both groups (#, *p* < 0.05; ## *p* < 0.01); no training effect for VO_2max_ or difference between groups for any measurement was observed.

**Figure 7 antioxidants-09-00879-f007:**
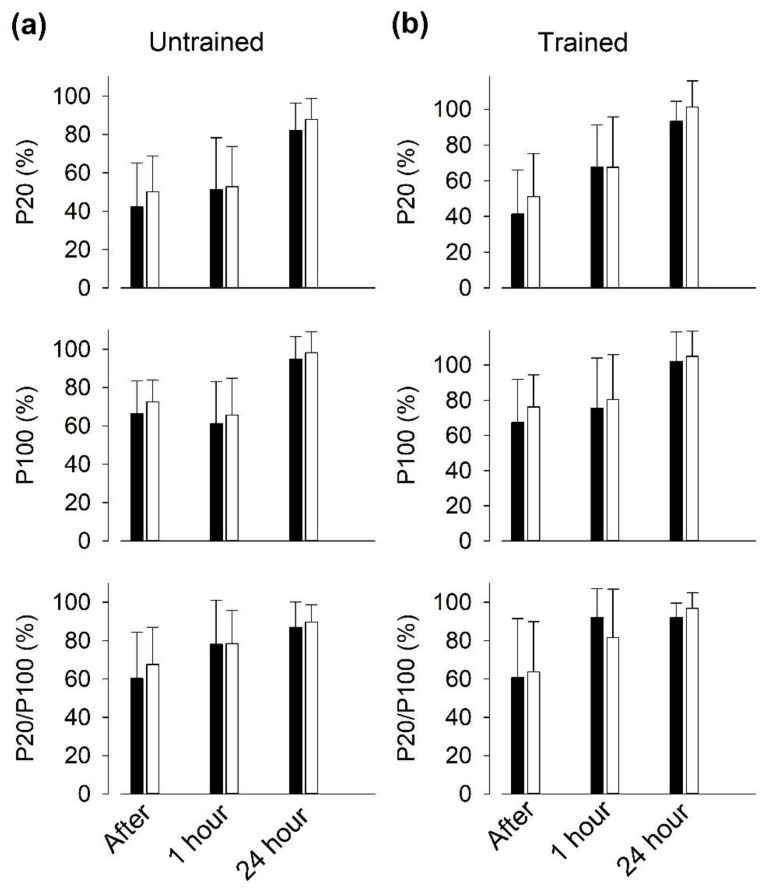
Recovery of electrically induced torque at low stimulation frequency after repeated all-out cycling bouts is slightly improved after three weeks of SIT. Isometric torque in response to supra-maximal electrical stimulation at 20 Hz (P20) and 100 Hz (P100), and the 20/100 Hz torque ratio presented relative to the value before cycling bouts, which was set to 100% in each subject. Data (mean ± SD) obtained at the start ((**a**), Untrained) and end ((**b**), Trained) of three weeks of SIT from contractions elicited immediately, 1 h and 24 h after six bouts of 30 s all-out cycling in individuals treated with vitamins (black bars; *n* = 10) or with placebo (white bars; *n* = 8). Two-way repeated measures ANOVA showed no significant difference between the two groups at any time.

**Table 1 antioxidants-09-00879-t001:** Characteristics of participants at the start of the study.

	Vitamin C + E (*n* = 10)	Placebo (*n* = 8)
Age (years)	67.0 ± 7.8	64.3 ± 6.0
Height (cm)	174.3 ± 4.0	179.6 ± 6.7
Body mass (kg)	81.9 ± 11.2	84.4 ± 12.1
BMI (kg m^−2^)	26.4 ± 2.7	26.0 ± 2.4
*Incremental cycling test*		
VO_2max_ (mL kg^−1^ min^−1^)	32.2 ± 7.1	34.5 ± 6.9
Maximal power (W kg^−1^)	2.82 ± 0.81	2.74 ± 0.51
Maximal heart rate (min^−1^)	154 ± 15	160 ± 19
RER_max_	1.15 ± 0.08	1,18 ± 0.05

Values are mean ± SD. BMI, body mass index; VO_2max_, maximal oxygen uptake; RER_max_, maximal respiratory exchange ratio.

**Table 2 antioxidants-09-00879-t002:** Primers used in quantitative real-time PCR analyses.

Protein	Abbreviation	Primer: FWD 5’- 3/ REV 5´- 3	NCBI Reference Sequence
***Inflammatory Mediators***			
Interleukin-1β	IL-1β	GGCATCCAGCTACGAATCTC/ GAACCAGCATCTTCCTCAGC	NM_000576.3
Interleukin-6	IL-6	GAAAGCAGCAAAGAGGCACT/ TTTCACCAGGCAAGTCTCCT	NM_14584.1
Interleukin-10	IL-10	CCAAGCTGAGAACCAAGACC/ GGGAAGAAATCGATGACAGC	NM_000572.3
Tumor necrosis factor-α	TNF-α	AACCTCCTCTCTGCCATCAA/ GGAAGACCCCTCCCAGATAG	NM_000594.4
Transforming growth factor-β1	TGF-β1	ACATTGACTTCCGCAAGGAC/ GTCCAGGCTCCAAATGTAGG	NM_000660.7
High mobility group box 1	HMGB1	CACCCAGATGCTTCAGTCAA/ TCCGCTTTTGCCATATCTTC	NM_002128.7
***ROS-Related Proteins***			
Glutathione peroxidase 1	GPX1	ACGATGTTGCCTGGAACTTT/ TCGATGTCAATGGTCTGGAA	NM_000581.4
Superoxide dismutase 1	SOD1	TGGCCGATGTGTCTATTGAA/ ACCTTTGCCCAAGTCATCTG	NM_000454.4
Superoxide dismutase 2	SOD2	GTTGGCCAAGGGAGATGTTA/ TAGGGCTGAGGTTTGTCCAG	NM_000636.4
Catalase 1	CAT1	CGTGCTGAATGAGGAACAGA/ TTGACCGCTTTCTTCTGGAT	NM_001752.4
NADPH oxidase 2	NOX2	AAATGGTGGCATGGATGATT/ TATTGACTCGGGCATTCACA	NM_000397.3
NADPH oxidase 4	NOX4	TGTTGGATGACTGGAAACCA/ AATCTGCAAACCAACGGAAG	NM_016931.5
***SR Ca^2+^-Handling Proteins***			
SR Ca^2+^-ATPase 1	SERCA1	TAAGAAGCTTGCCCTCCGTA/ CAGACATCTGGTTGGTGGTG	NM_004320.4
SR Ca^2+^-ATPase 2a	SERCA2a	CTGAAGAAAGCCGAGATTGG/ GCCACAATGGTGGAGAAGTT	NM_170665.4
Calsequestrin 1	CSQ1	TCCCATACTGGGAGAAGACG/ TCCTCCTCATCGTCCATTTC	NM_001231.5
Calsequestrin 2	CSQ2	GACAAAGGGGTTGCAAAGAA/ CTCCACCAGCTCCTCTTCTG	NM_001232.3
Dihydropyridine receptor	DHRP	ACTGTATTGCCTGGGTGGAG/ GCTTGATCAGCCTCATGACA	U09784.1
Ryanodine receptor 1, C-terminal	RyR1-C	ACAGGGTGGTCTTCGACATC/ GTCTCGGAGCTCACCAAAAG	NM_000540.2
Ryanodine receptor 1, N-terminal	RyR1-N	TGCTGCAGACAAACCTCATC/ ATTTGCTGTACTGCGTGGTG	NM_000540.2
***Mitochondria-Related Proteins***			
Dynamin 1-like protein	DNM1L	AAATCGTCGTAGTGGGAACG/ CGGGTGACAATTCCAGTACC	NM_012062.4
Mitofusin 2	MFN2	GGCCAAACATCTTCATCCTG/ CTGGTACAACGCTCCATGTG	NM_014874.3
Mitochondrial E3 ubiquitin protein ligase 1	MUL1	GAGAAGTTCCACCCCTCGAT/ TCAGCATCTCCTCGGTCTCT	NM_024544.3
Mitochondrial elongation factor 2	MIEF2	GACTTCCTCCTGGCCAATG/ TGGCCCTGTCAATGAACC	NM_139162.4
Mitochondrial dynamin-like GTPase	OPA1	GGGTTGTTGTGGTTGGAGAT/ GTCATCATCTCCCCAGATCC	NM_015560.2
Peroxisome proliferator-activated receptor α	PPARα	CCTTGCAGCACAAGAAAACA/ CTGCTTCGTCGTCAAAAACA	NM_013261.5
Peroxisome proliferator-activated receptor δ	PPARδ	ACTGAGTTCGCCAAGAGCATC/ ACGCCATACTTGAGAAGGGTAA	NM_138712.4
***House-Keeping Gene***			
Hypoxanthine-guanine Phosphoribosyltransferase	HPRT	GAAAAGGACCCCACGAAGTGT/ AGTCAAGGGCATATCCTACAA	NM_000194

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
