# Peer review of "Vitamin C and E Treatment Blunts Sprint Interval Training–Induced Changes in Inflammatory Mediator-, Calcium-, and Mitochondria-Related Signaling in Recreationally Active Elderly Humans"

_antioxidants, 2020, doi:10.3390/antiox9090879_

Round 1

Reviewer 1 Report

Wyckelsma et al. investigate the effect of Vitamin C and E supplementation on acute sprint exercise inflammatory mediator-, calcium-3 , and mitochondria-related signaling responses pre and post training in elderly men. In general they provide evidence that these supplements may be altering acute exercise responses to a greater extend following training, without greatly effecting training-induced performance responses. While I believe the study is well designed and that such results will be of interested to the sport and exercise science community, the message of the paper could be improved with further analysis and clarification of results.      

My comments/suggestions are as follows:

  1. Consider removing the hypothesis ‘we hypothesized that the positive effects on endurance would be less clear in the present elderly participants than in previous studies on young individuals’. While this is possible to speculate based on previous literature, to address it systematically a young groups would be included in the study. This study does not appear to be designed to address this hypothesis.

  1. Vitamin C and E use are consistent with what has been previously used in similar research models, and likely consistent with what supplements the public might be using. However, both vitamins play a wider role in cellular function than simply acting as antioxidant, can in some circumstances be pro-oxidants. Do you have any data from this cohort to suggest they are having antioxidant effects? If not it is suggested that narrative be reframed to assess the effect of these vitamins on SIT response in your cohort/intervention, rather than antioxidants or ROS signalling. Certainly the antioxidant potential of these should still be addressed, but the direct assertion that results are causatively linked to alterations in ROS (without evidence) may need re-considering.

  1. Were participants in a fed or fasted state when biopsies were taken? If fed, was pre-biopsy diet controlled. Similar question applies to exercise performance tests.

  1. Could you please explain why t-tests were used for training differences rather than ANOVA (which was used all other measures). A repeated measures ANOVA or mixed linear with post-hocs if there is a main effect is perhaps more appropriate?

  1. The n’s (3-4) for figure 1 are less than half that of other figures. Presumably these measures were not taken in the other 4-5 participants per group. I believes inferences from this very small n to larger group performance is limited and thus this figure should be removed or moved to supplementary data. I don’t think removing this data would in any way weaken the manuscript.

  1. Figure 2 and other gene/protein data. Data are presented as individual fold from pre exercise. While I believe this is a reasonable way to present the data to assess changes is response to exercise, can the authors also present a comparison of pre-exercise gene expression between training status and supplement groups (and/or provide detadeta CT data relative to pre training placebo). This would allow assessment of whether supplements may be effecting basal expression of these genes, and whether there was a training effect on basal expression (also see next comment for importance).

  1. Line 258: ‘highly significant general group difference (P < 0.001)’. Could this please be clarified, and similar statements also clarified- is this a supplement effect, training effect and/or interaction effect and for which genes. This is important for interpretation of the data. With reference to figure 3, it appears few of these genes changed with acute exercise, but there is an antioxidant effect? Is this correct, and did difference in pre-exercise levels effect extent of exercise response?    

  1. Was there a reason what 24h samples were not analysed for inflammatory or ROS-related genes?

  1. Line 295. ‘The full-length RyR1 expression was significantly decreased 24 hours after the first SIT session in both groups (P < 0.05 with two-way repeated measures ANOVA).’ Is this referring to figure 4a? If so, please provide a quantification of blots to justify this statement. Why is the n lower in these figures?  

  1. Line 296. ‘We also observed a decrease in average full-length RyR1 expression 24 hours after the last SIT session, but the variability was large and the decrease did not reach statistical significance’. Consider rephrasing to there was no difference if this is what the data shows. Please show quantification and ideally a few more representative blots to make this point.

  1. Line 299-304. To justify this statement please show individual data points in quantification and lines linking an individual’s response. It is also highly recommended that for all data individual data points are shown rather than group means (as done form performance data).

  1. Line 308: ‘ROS-related proteins’. Do you mean genes? and probably antioxidants and NOX’s rather than ROS.

  1. Figure 6a. This data suggest that antioxidant blunted training effects on mitochondrial-related mRNA, however for this to determined can the authors clarify whether there was a interaction effect between exercise training and supplementation?

  1. Why do you think that supplementation had differential effects on acute exercise response pre-and post-training? This seems to be a major finding is not really addressed. Could this simply be the result of being supplemented longer? Presenting absolute gene and protein data and/or making pre-exercise comparisons across groups and training status may help address this question.

  1. Section 4.3 (line 485). Data to support points 1-2 (see earlier comments) is not presented, please revise accordingly. The second two points are very speculative and should be considered carefully. Also consider removing dot points.

  1. It is generally stated that the cell signalling data does not support training-induced changes in performance data. Which is true to an extent, however would be further supported by comparing whether supplement group showed similar responses in mitochondrial (or other) gene expression to training per se as placebo group. ie did the training program increased pre-exercise mitochondrial genes and was this altered by supplement?

  1. Based on the comments above, conclusions may need to be amended.

Author Response

Response to Reviewer 1:

  1. Consider removing the hypothesis ‘we hypothesized that the positive effects on endurance would be less clear in the present elderly participants than in previous studies on young individuals’. While this is possible to speculate based on previous literature, to address it systematically a young groups would be included in the study. This study does not appear to be designed to address this hypothesis.

Response: The hypothesis has been removed.

  1. Vitamin C and E use are consistent with what has been previously used in similar research models, and likely consistent with what supplements the public might be using. However, both vitamins play a wider role in cellular function than simply acting as antioxidant, can in some circumstances be pro-oxidants. Do you have any data from this cohort to suggest they are having antioxidant effects? If not it is suggested that narrative be reframed to assess the effect of these vitamins on SIT response in your cohort/intervention, rather than antioxidants or ROS signalling. Certainly the antioxidant potential of these should still be addressed, but the direct assertion that results are causatively linked to alterations in ROS (without evidence) may need re-considering.

Response: The previously named antioxidant group is now referred to as the vitamin group throughout the manuscript.

  1. Were participants in a fed or fasted state when biopsies were taken? If fed, was pre-biopsy diet controlled. Similar question applies to exercise performance tests.

Response: The subjects were not fasted but were instructed to maintain their regular diet and no food intake for at least two hours before the first and the last SIT sessions as well as the pre- and post-training testing sessions. This information is now added in Materials and Methods (section 2.1).

  1. Could you please explain why t-tests were used for training differences rather than ANOVA (which was used all other measures). A repeated measures ANOVA or mixed linear with post-hocs if there is a main effect is perhaps more appropriate?

Response: As suggested by the reviewer, the statistical test has been changed to two-way repeated measures ANOVA for all analyses.

  1. The n’s (3-4) for figure 1 are less than half that of other figures. Presumably these measures were not taken in the other 4-5 participants per group. I believes inferences from this very small n to larger group performance is limited and thus this figure should be removed or moved to supplementary data. I don’t think removing this data would in any way weaken the manuscript.

Response: Due to technical problems, measurements were only obtained in a subgroup of participants. As suggested, Figure 1 has been moved to Supplementary Material (Figure S1).

  1. Figure 2 and other gene/protein data. Data are presented as individual fold from pre exercise. While I believe this is a reasonable way to present the data to assess changes is response to exercise, can the authors also present a comparison of pre-exercise gene expression between training status and supplement groups (and/or provide detadeta CT data relative to pre training placebo). This would allow assessment of whether supplements may be effecting basal expression of these genes, and whether there was a training effect on basal expression (also see next comment for importance).

Response: We thank the reviewer for this suggestion. Pre-exercise values in the untrained and trained state for each participant are now presented as Supplementary Figures. Some interesting (and by us overlooked) findings appeared and these are now included in the Results and Discussions sections. Moreover, we have added individual values to all Figures showing relative mRNA and protein expressions.

  1. Line 258: ‘highly significant general group difference (P < 0.001)’. Could this please be clarified, and similar statements also clarified- is this a supplement effect, training effect and/or interaction effect and for which genes. This is important for interpretation of the data. With reference to figure 3, it appears few of these genes changed with acute exercise, but there is an antioxidant effect? Is this correct, and did difference in pre-exercise levels effect extent of exercise response?    

Response: The reported group effects refer to conditions where the two-way repeated ANOVA test shows an overall difference between the vitamin and the placebo groups. This has now been clarified in the Materials and Methods (section 2.7): “Two-way repeated measures ANOVA was used to identify a general group difference between vitamin- and placebo-treated participants. The Holm-Sidak post hoc test was used to identify differences in the expression of individual genes or proteins between vitamin- and placebo treated participants.

  1. Was there a reason what 24h samples were not analysed for inflammatory or ROS-related genes?

Response: Due to limited amount of muscle biopsy tissue, we had to prioritize what to analyse at the different time points. At 24h after exercise, a large amount of the available tissue was used for the RyR1 Western blots (see below) and hence gene expression analysis could not be performed. This is now explained in Materials and Methods (section 2.3) together with the rationale for looking at changes in expression of genes at the earlier and proteins at the later time point.

  1. Line 295. ‘The full-length RyR1 expression was significantly decreased 24 hours after the first SIT session in both groups (P < 0.05 with two-way repeated measures ANOVA).’ Is this referring to figure 4a? If so, please provide a quantification of blots to justify this statement. Why is the n lower in these figures?  

Response: Yes, it refers to the original Figure 4a and b (now Figure 3a and b). Quantifications of the blots are shown in Figure 3b and individual values are now added. We now also indicate the statistically significant decrease in full-length RyR1 expression in the untrained state in Figure 3b. Due to its large size, RyR1 is often difficult to analyse with Western blotting and combined with limited amount of tissue, this resulted in reliable measurements from fewer participants than for other protein measurements.

  1. Line 296. ‘We also observed a decrease in average full-length RyR1 expression 24 hours after the last SIT session, but the variability was large and the decrease did not reach statistical significance’. Consider rephrasing to there was no difference if this is what the data shows. Please show quantification and ideally a few more representative blots to make this point.

Response: The statement has been rephrased as suggested. We hope that the addition of individual values in Figure 3b makes it easier for the readers to appraise our analysis at the individual participant level. All Western blots have been submitted to the journal.

  1. Line 299-304. To justify this statement please show individual data points in quantification and lines linking an individual’s response. It is also highly recommended that for all data individual data points are shown rather than group means (as done form performance data).

Response: The statement has been reworded and individual data points are now shown.

  1. Line 308: ‘ROS-related proteins’. Do you mean genes? and probably antioxidants and NOX’s rather than ROS.

Response: The text has been reworded to clarify that we refer to mRNA expression of ROS-related proteins. We consider antioxidants and NOX´s to be ROS-related proteins and would like to keep this general naming. The risk of misunderstandings should be minimal since the text and Figure 2 clearly show which proteins we refer to.

  1. Figure 6a. This data suggest that antioxidant blunted training effects on mitochondrial-related mRNA, however for this to determined can the authors clarify whether there was a interaction effect between exercise training and supplementation?

Response: We have revised the text to make it clear that the mRNA response to the SIT session in the trained state was markedly larger in the placebo than in the vitamin group.

  1. Why do you think that supplementation had differential effects on acute exercise response pre-and post-training? This seems to be a major finding is not really addressed. Could this simply be the result of being supplemented longer? Presenting absolute gene and protein data and/or making pre-exercise comparisons across groups and training status may help address this question.

Response: Individual and pre-exercise data have been added. We are intrigued by the different acute responses in the untrained and trained states and believe that this ought to be addressed in future studies. This is now briefly discussed in the first section of the Discussion.

  1. Section 4.3 (line 485). Data to support points 1-2 (see earlier comments) is not presented, please revise accordingly. The second two points are very speculative and should be considered carefully. Also consider removing dot points.

Response: We have removed the dot point formatting and replaced it by numbered standardly formatted paragraphs. We believe that data to support points 1 and 2 are now presented (see points above). In addition, the new assessment of pre-exercise mRNA expression (see point 6) revealed decreased RyR1 mRNA expression in the trained state in vitamin-treated subjects, which also shows vitamin-dependent effects on RyR1 and this finding has been as a new point (iii). Changes in RyR1 occur in various situations and have received increasing attention in recent years. Mechanisms underlying these changes are not fully understood and debated. The previous point 3 (now point iv) refers to a novel mechanism of RyR1 modification. This point is now reworded to highlight that it is speculative, but we want to keep it since it may promote future studies and thereby increase our understanding of RyR1 modifications and their underlying mechanisms. We agree that the previous point 4 was too speculative and it has been deleted.

  1. It is generally stated that the cell signalling data does not support training-induced changes in performance data. Which is true to an extent, however would be further supported by comparing whether supplement group showed similar responses in mitochondrial (or other) gene expression to training per se as placebo group. ie did the training program increased pre-exercise mitochondrial genes and was this altered by supplement?

Response: Pre-exercise data of mRNA for mitochondria-related proteins are now presented in Figure S6. There were no consistent differences in pre-exercise gene expression between the two groups. This issue is now briefly discussed in the first section of the Discussion.

  1. Based on the comments above, conclusions may need to be amended.

Response: The Conclusions section has been amended.

Reviewer 2 Report

The authors examined the effects of antioxidant on exercise-induced skeletal muscle adaptations in elderly humans. This is a well-written manuscript describing a well conducted complex study. The only concern is the timing of muscle biopsy. What was the reason to take the samples at 1h after exercise? Didn't it affect the result? Is there any possibility of missing the peak of most genes?

Author Response

Response to Reviewer 2.

The authors examined the effects of antioxidant on exercise-induced skeletal muscle adaptations in elderly humans. This is a well-written manuscript describing a well conducted complex study. The only concern is the timing of muscle biopsy. What was the reason to take the samples at 1h after exercise? Didn't it affect the result? Is there any possibility of missing the peak of most genes?

Response: We agree with the reviewer and in retrospect it might have been better to take the first post-exercise biopsy at 3-4 hours. However, we wanted to avoid the risk of missing short-lasting peaks in mRNA expression. Even if we probably missed the peak expression of many of our studied genes, we observed a distinct pattern of post-exercise changes in the trained state. These issues are now discussed in the first section of the Discussion.

Round 2

Reviewer 1 Report

The authors have adequately addressed my prior concerns. Congratulations on a much improved manuscript.